# Event2Vec: Processing Neuromorphic Events Directly by Representations in Vector Space

**Wei Fang** [1]  **Priyadarshini Panda** [2]

## Abstract

Neuromorphic event cameras possess superior temporal resolution, power efficiency, and dynamic range compared to traditional cameras. However, their asynchronous and sparse data format poses a significant challenge for conventional deep learning methods. Most existing methods either densify events into frames, sacrificing their sparse asynchronous nature, or use irregular models that are less compatible with GPU acceleration. Inspired by word-to-vector models, we propose event2vec, a novel representation that allows Transformers to process events directly. We demonstrate the effectiveness of event2vec on the DVS Gesture, ASL-DVS, and DVS-Lip benchmarks, showing that event2vec is remarkably parameter-efficient, features high throughput and low latency, and achieves high accuracy even with an extremely low number of events or low spatial resolutions. These results show that sparse asynchronous event data can be directly integrated into high-throughput Transformer architectures, offering an efficient paradigm for real-time neuromorphic vision. The code is provided at https://github.com/Intelligent-Computing-Lab-Panda/event2vec.

## 1. Introduction

Neuromorphic computing is an emerging research field that seeks to develop the next generation of artificial intelligence by emulating the brain's principles (Mead, 1990). A significant advancement stemming from this paradigm is the event camera, a sensor inspired by the biological retina (Gallego

[1]Electrical & Computer Engineering, Yale University [2]Electrical & Computer Engineering, University of Southern California. Correspondence to: Priyadarshini Panda <priya.panda@usc.edu>.

*Proceedings of the 43$^{rd}$ International Conference on Machine Learning*, Seoul, South Korea. PMLR 306, 2026. Copyright 2026 by the author(s).

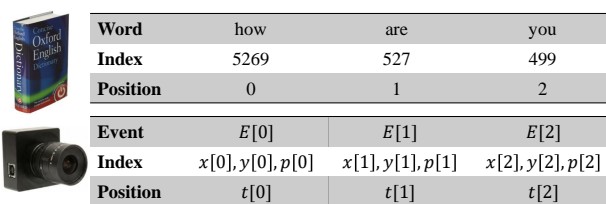

| Word | how | are | you |
|---|---|---|---|
| **Index** | 5269 | 527 | 499 |
| **Position** | 0 | 1 | 2 |

| Event | $E[0]$ | $E[1]$ | $E[2]$ |
|---|---|---|---|
| **Index** | $x[0], y[0], p[0]$ | $x[1], y[1], p[1]$ | $x[2], y[2], p[2]$ |
| **Position** | $t[0]$ | $t[1]$ | $t[2]$ |

*Figure 1.* Conceptual analogy between words and events.

et al., 2022). Prominent examples include the Dynamic Vision Sensor (DVS) (Lichtsteiner et al., 2008) and the Asynchronous Time-based Image Sensor (ATIS) (Posch et al., 2011). Unlike traditional cameras that capture synchronous frames, event cameras operate asynchronously, generating events in response to per-pixel brightness changes. This operational principle endows them with exceptionally high temporal resolution (on the order of microseconds), low power consumption, and a High Dynamic Range (HDR) exceeding 120 dB. This asynchronous operation results in a sparse stream of events, typically encoded in the Address-Event Representation (AER) format. An event is represented as a tuple $(x, y, t, p)$, composed of the pixel's spatial coordinates $(x, y)$, a timestamp $t$, and a binary polarity $p$ that indicates the direction of the brightness change.

Most contemporary deep learning models are designed to operate on dense, regularly structured, multi-dimensional tensors. This regular paradigm is foundational to mainstream deep learning (LeCun et al., 2015) and is ubiquitously employed in modern scientific computing and machine learning frameworks, including NumPy (Harris et al., 2020), TensorFlow (Abadi et al., 2016), and PyTorch (Paszke et al., 2019). Consequently, the sparse and asynchronous nature of event streams in the AER format is fundamentally incompatible with these regular methods. To address this disparity, substantial research efforts have been devoted to converting events to dense representations or designing new data and network structures to process the irregular events directly.

Existing methods primarily address the challenge of event encoding: how to effectively extract information from events and represent it for processing by neural networks. This challenge is analogous to word encoding in natural language processing, a problem successfully addressed by word-to-vector (word2vec) (Mikolov et al., 2013). The word2vec

model embeds each word into a fixed-length vector, enabling the relationships between words to be represented by mathematical operations between vectors. This vector representation approach is highly compatible with deep learning architectures and has become a foundational component of modern Natural Language Processing (NLP) models (Devlin et al., 2019; Brown et al., 2020). As illustrated in Figure 1, we identify numerous parallels between words and events. The key similarities are as follows:

(1) **Each element is a composite of an index and a position.** In NLP, each word is assigned a unique index from a vocabulary, a conversion handled by a tokenizer; the indices in Figure 1, for instance, are generated by the Llama-3 tokenizer (Grattafiori et al., 2024). A word's position is its sequential location within the sentence (e.g., the word "how" is at position 0 in "how are you"). Similarly, an event's index is its spatial address, represented by the tuple $(x, y, p)$. Crucially, its position is not the sequence number, but its timestamp $t$, which marks its precise temporal location in the event stream.

(2) **The set of possible indices is finite.** The vocabulary of a language, which forms the dictionary used in NLP, is finite. Likewise, an event camera has a limited set of possible event indices, defined by its sensor's properties. For example, a DVS128 camera has $2 \times 128 \times 128$ unique indices, corresponding to two polarities across a $128 \times 128$ spatial resolution.

(3) **The sequence exhibits a natural ordering.** Words in a sentence are arranged in a specific sequence that dictates meaning. Analogously, events are naturally ordered by their timestamps, reflecting the chronological progression of captured changes. This inherent temporal order is a key characteristic that distinguishes event data from unordered data structures like point clouds.

(4) **The meaning of an element is determined by its context.** A word can be polysemous; for instance, "transformer" can refer to a neural network architecture or a character in an animated series; its specific meaning is disambiguated by the surrounding text. An individual event merely indicates a brightness change at a specific pixel and time, conveying little information in isolation. However, when viewed within a spatiotemporal stream, a sequence of events can delineate an object's contour, thus giving a single event a higher-level meaning, such as being part of an edge. Therefore, the significance of an event is also fundamentally context-dependent.

Inspired by word2vec, we propose event-to-vector (event2vec), an efficient spatiotemporal representation for asynchronous events. Our contributions are as follows:

(1) By embedding events into a vector space, our method natively handles the sparse nature of the input stream, avoiding dense intermediate representations like event frames. This allows for efficient, GPU-accelerated processing with modern network architectures.

(2) We propose a parametric spatial embedding and a convolution-based temporal embedding method to capture neighborhood similarity—a characteristic that is critical for accuracy but difficult for a standard embedding layer (a lookup table) to learn.

(3) We validated our method on three widely used classification benchmarks: DVS Gesture (Amir et al., 2017), ASL-DVS (Bi et al., 2019), and DVS-Lip (Tan et al., 2022). It achieved competitive accuracy while demonstrating remarkable parameter efficiency, throughput, latency, as well as robustness against a low number of events or low spatial resolutions.

## 2. Related Work

### 2.1. Dense Representations and Processing of Events

Dense representations, derived from raw event streams, are fully compatible with conventional deep learning methods. This is typically achieved by integrating events along the time axis to form dense 3D or 4D tensors, such as event frames (Liu & Delbruck, 2018), multi-channel images (Barchid et al., 2022), voxel grids (Bardow et al., 2016), volumetric cubes (Cordone et al., 2022), and patches (Sabater et al., 2023; Peng et al., 2023). Specifically, event-to-frame methods accumulate events within discrete time intervals. The resulting frames can then be processed directly by standard neural networks. However, a significant drawback of these methods is the degradation of the high temporal resolution inherent to event data. This occurs because individual event timestamps are aggregated or quantized during the conversion process. Furthermore, transforming the data into a dense representation negates the inherent spatial sparsity of events. For instance, the generated frames often contain a substantial number of zero-valued pixels. These pixels, while carrying no information, still incur significant memory and computational overhead. While many methods use timestamps implicitly to define the integration interval, some approaches explicitly leverage them to generate temporal weights (Zhu et al., 2019; Gehrig et al., 2019). Finally, the conversion process itself can be computationally intensive, introducing considerable latency that is often prohibitive for real-time applications (Rebecq et al., 2019; Gallego et al., 2022).

### 2.2. Irregular Representations and Processing of Events

Conversely, methods for processing irregular representations aim to preserve the inherent sparsity and asynchronicity of event data. This category includes Spiking Neural Networks (SNNs) (Maass, 1997; Roy et al., 2019), Sparse Con-

volutional Networks (Sparse CNNs) (Messikommer et al., 2020; Santambrogio et al., 2024), Graph Neural Networks (GNNs) (Bi et al., 2019; Schaefer et al., 2022), and point-based methods (Yang et al., 2019; Sekikawa et al., 2019; Lin et al., 2023; Ren et al., 2025).

When deployed on neuromorphic hardware (Merolla et al., 2014; Davies et al., 2018), SNNs can process events in a naturally asynchronous, event-driven manner. However, on standard hardware, GPU-based simulations of SNNs produce dense tensor outputs, as the hardware necessitates synchronous processing with discrete time steps. Consequently, training SNNs on GPUs typically occurs in a synchronous fashion, leading to an unavoidable performance gap between synchronous training and asynchronous inference (Yao et al., 2024; Du et al., 2025). Moreover, the reliance on backpropagation-through-time renders the training process slow and memory-intensive. Sparse CNNs leverage the inherent sparsity of event data, achieving a theoretically low number of floating-point operations (FLOPs). Nevertheless, GPU architectures are not optimized for the dynamic computations and unstructured memory access patterns required for efficient sparse acceleration. Consequently, similar to SNNs, Sparse CNNs fail to fully exploit the massive parallel processing capabilities of GPUs.

Event-based GNNs construct graphs from incoming events, an approach that effectively preserves the spatiotemporal relationships between them. Since empty regions with no event activity do not generate graph nodes, the data's sparsity is well-utilized. Their main disadvantage lies in the need for careful hyperparameter tuning, such as the event downsampling rate and neighborhood radius for graph construction. Additionally, functioning as low-pass filters (Nt & Maehara, 2019), GNNs are susceptible to the over-smoothing problem (Zhou et al., 2020), which limits their ability to form deep architectures comparable to modern CNNs and Transformers (Vaswani et al., 2017). Point-based methods treat events from event cameras as analogous to point clouds from Light Detection and Ranging (LiDAR) sensors. A fundamental limitation of most point cloud models is their permutation invariance, which necessitates treating the input as an unordered set. Consequently, the event timestamp is typically relegated to being an additional positional coordinate, thereby discarding the crucial causal ordering of events. To manage the data volume, these methods often employ classic point cloud preprocessing techniques like farthest point sampling, which further increases latency.

## 3. Methods

### 3.1. Representing Events in a Vector Space

Leveraging the strong analogy between words and events, we propose a method for representing events within a vector space, which we term event-to-vector (event2vec). An event, generated by a camera with a spatial resolution of $H \times W$, is represented as a tuple $(x, y, t, p)$. For our embedding, we treat the triplet $(x, y, p)$ as the spatial-polarity coordinate and the timestamp $t$ as the temporal coordinate. The general formulation for the event2vec embedding is defined as:

$$\mathbf{v} = \mathbf{v}_s + \mathbf{v}_t = \text{Embed}_s(x, y, p) + \text{Embed}_t(\Delta t), \quad (1)$$

where $\mathbf{v} \in \mathbb{R}^D$ is the resulting $D$-dimensional embedding vector, $\mathbf{v}_s = \text{Embed}_s(x, y, p) \in \mathbb{R}^D$ is the spatial-polarity embedding vector, and $\mathbf{v}_t = \text{Embed}_t(\Delta t) \in \mathbb{R}^D$ is the temporal embedding vector. As shown in Eq. 1, this method fuses spatial and temporal information through addition. This additive fusion strategy is directly inspired by the positional encoding mechanism prevalent in Transformers.

### 3.2. Spatial Embedding

A straightforward approach for the spatial embedding module is to adapt the standard embedding layer from NLP, which is efficiently implemented as a lookup table:

$$\mathbf{v}_s = \text{Embed}_s(x, y, p) = \mathbf{W}_s[p \cdot H \cdot W + y \cdot W + x] \quad (2)$$

where $\mathbf{W}_s \in \mathbb{R}^{(P \cdot H \cdot W) \times D}$ is the learnable embedding matrix and $D$ is the embedding size. This method maps each unique spatial-polarity coordinate to a distinct row index in the embedding matrix $\mathbf{W}_s$. This lookup-based spatial embedding is analogous to the lookup-table realization of EventNet's spatial mapping (Sekikawa et al., 2019), since both operate over the finite $2HW$ spatial-polarity address space; we discuss this connection together with the temporal component in Sec. 3.5.

However, this standard embedding layer imposes no inductive bias on the relationship between indices, compelling the model to learn all spatial relationships from data alone. In a tokenizer, a word's index is a non-semantic identifier, the assignment of which is primarily determined by the word's frequency in the training corpus. Consequently, the words at indices $i$ and $i + 1$ share no inherent semantic similarity. This assumption does not hold for event coordinates. Images are continuous two-dimensional functions (Gonzalez, 2009). Spatially adjacent pixels are known to exhibit strong correlation. Therefore, an effective spatial embedding should incorporate this locality bias, ensuring that events with close coordinates yield similar embedding vectors:

$$\text{Embed}_s(x + \Delta x, y + \Delta y, p) - \text{Embed}_s(x, y, p) \approx \mathbf{0} \quad (3)$$

for small coordinate perturbations $(\Delta x, \Delta y)$.

The standard embedding in Eq. 2 fails to account for this crucial spatial relationship, which can impede the learning process. To solve this issue, we propose an elegant parametric embedding implemented by a neural network $\phi$ that

directly maps each normalized coordinate triplet $(x, y, p)$ to its embedding. Conceptually, this parametric embedding can also be viewed as generating an embedding matrix $\mathbf{W}_\phi$ by evaluating $\phi$ over all possible spatial-polarity coordinates. To systematically enumerate all spatial-polarity coordinates within a $P \times H \times W$ volume (where $P = 2$ represents the two polarities), we can first establish a linear index sequence $\mathbf{c} = [0, 1, \dots, P \cdot H \cdot W - 1]$. This sequence is then decomposed into three probe tensors, $\mathbf{x}_c$, $\mathbf{y}_c$, and $\mathbf{p}_c$, which correspond to the coordinates along the width, height, and polarity dimensions, respectively. The transformation is defined as follows: $\mathbf{x}_c = \mathbf{c} \pmod{W}, \mathbf{y}_c = \left\lfloor \frac{\mathbf{c}}{W} \right\rfloor \pmod{H}, \mathbf{p}_c = \left\lfloor \frac{\mathbf{c}}{WH} \right\rfloor$. Before being passed to $\phi$, these coordinate tensors are normalized to $[-1, 1]$, following the implementation:

$$\bar{\mathbf{x}}_c = \frac{2\mathbf{x}_c}{W-1} - 1, \quad \bar{\mathbf{y}}_c = \frac{2\mathbf{y}_c}{H-1} - 1, \quad \bar{\mathbf{p}}_c = \frac{2\mathbf{p}_c}{P-1} - 1. \quad (4)$$

Finally, these normalized probe tensors are passed through $\phi$, which gives the conceptual embedding matrix $\mathbf{W}_\phi = \phi(\bar{\mathbf{x}}_c, \bar{\mathbf{y}}_c, \bar{\mathbf{p}}_c)$. This parametrically generated matrix is equivalent to evaluating $\phi$ directly on any given raw event coordinate $(x, y, p)$ after the same normalization to $(\bar{x}, \bar{y}, \bar{p})$:

$$\mathbf{W}_\phi[p \cdot H \cdot W + y \cdot W + x] = \phi(\bar{x}, \bar{y}, \bar{p}). \quad (5)$$

Crucially, the parametric network $\phi$ is designed to be a continuous and differentiable function. This property allows us to formally analyze the relationship between neighboring embeddings using a first-order Taylor series expansion:

$$\phi(x + \Delta x, y + \Delta y, p) - \phi(x, y, p)$$
$$= J_\phi^{x,y}(x, y, p) \begin{bmatrix} \Delta x \\ \Delta y \end{bmatrix} + o\left(\|(\Delta x, \Delta y)\|\right), \quad (6)$$

where $J_\phi^{x,y}(x, y, p)$ denotes the Jacobian of $\phi$ with respect to the spatial coordinates $(x, y)$. As Eq. 6 illustrates, for small perturbations $(\Delta x, \Delta y)$, the difference between the embeddings is approximated by the matrix-vector product between this Jacobian and the perturbation vector. Consequently, as the perturbations approach zero, this difference vector also approaches zero. In this manner, a continuous parametric network $\phi$ inherently embeds the desired neighborhood semantics, or spatial inductive bias, directly into the embedding matrix. This approach elegantly satisfies the condition outlined in Eq. 3.

### 3.3. Temporal Embedding

Timestamps, which denote the occurrence time of events, serve a function analogous to positional indices in a sentence. In modern NLP models, relative positional encoding methods (Press et al., 2021; Su et al., 2024) are increasingly favored over absolute methods, such as sinusoidal encoding (Vaswani et al., 2017) or learnable absolute positional embeddings (Devlin et al., 2019).

However, directly applying these relative positional encoding techniques to event timestamps is ill-suited. Such methods are fundamentally designed for discrete and uniformly spaced indices, whereas event timestamps are continuous and inherently non-uniform. To address this discrepancy, we propose learning the temporal embedding directly from the differences between consecutive timestamps.

Specifically, the temporal embedding module is implemented as a stack of convolutional layers that takes the sequence of the first-order temporal differences of normalized timestamps as input. Let $\tilde{\mathbf{t}} = \mathbf{t} / \max(\mathbf{t})$ for each event stream. We use $\Delta \mathbf{t} = [0, \tilde{\mathbf{t}}[1] - \tilde{\mathbf{t}}[0], \tilde{\mathbf{t}}[2] - \tilde{\mathbf{t}}[1], \dots, \tilde{\mathbf{t}}[L-1] - \tilde{\mathbf{t}}[L-2]]$, where $L$ is the number of events and the initial 0 aligns the interval sequence with the event sequence. This design offers several advantages:

(1) **Time-Shift Invariance:** By operating on relative temporal differences, the embedding becomes inherently invariant to absolute shifts in time.

(2) **Contextual Consistency:** The convolutional operations allow the temporal embedding for an event to be influenced by the timing of its immediate neighbors, thereby reinforcing the principle of neighborhood semantics in the time domain. On the other hand, the occurrence of individual events may contain a certain amount of noise, and convolution is applied to achieve the effect of local smoothing and noise reduction.

(3) **Optimization Efficiency and Inductive Bias:** Providing $\Delta \mathbf{t}$ as input serves as a form of temporal 'preconditioning', aligning with the principle of residual learning (He et al., 2016). While a network could theoretically infer intervals from absolute timestamps $\mathbf{t}$, explicitly modeling $\Delta \mathbf{t}$ reduces the optimization burden by providing a direct representation of event velocity. Since the convolution operation essentially performs a weighted summation, it is mathematically congruent with differential inputs $\Delta \mathbf{t}$. The summation of consecutive time differences possesses a clear physical interpretation: it represents the accumulated duration of a local event window, enabling the network to directly measure the local event density.

### 3.4. Event Sampling and Aggregation

Raw event streams often contain an extremely large number of events, with sequence lengths exhibiting substantial variance. Furthermore, deep learning frameworks typically process data in batches, which requires that all tensors within a single batch have uniform dimensions. Consequently, it is necessary to sample or aggregate events from each stream to a fixed-length sequence of size $L$.

In this paper, we primarily use two methods. The first is **uniform random sampling**. We find that this straightforward

method works well in most cases and is extremely computationally efficient. However, a significant limitation of random sampling is the substantial information loss incurred by discarding the majority of the events, leading to suboptimal accuracy in complex tasks. Our second method addresses this by leveraging **K-Means clustering** to aggregate the entire event stream into $L$ representative clusters. Specifically, the clustering process is performed independently on the two event polarities to preserve their distinct information channels. Furthermore, we compute an intensity factor, $\rho$, equal to the number of raw events belonging to that cluster. This intensity factor then modulates the corresponding event token, effectively weighting the representation by its event density.

To reduce the latency of running the K-Means clustering algorithm during inference, we propose a GPU-based **batched K-Means++ algorithm**. This method approximates the step-by-step iteration of K-Means++ initialization (Arthur & Vassilvitskii, 2007) via multi-step batch computation. Meanwhile, it incrementally updates the nearest-center distances using only the newly sampled batch of centers, and achieves an efficient GPU-based implementation with PyTorch. Details can be found in Appendix A.1.

### 3.5. The Formulation of Event2Vec

In summary, the final event2vec representation for a sequence of $L$ events is a tensor $\mathbf{V} \in \mathbb{R}^{L \times D}$. The embedding for the $i$-th event in this sequence, $\mathbf{V}[i]$, is formulated as:

$$
\begin{aligned}
\mathbf{V}[i] = (\log(\boldsymbol{\rho}[i]) + 1)\cdot \\
\left( \text{Embed}_s(\mathbf{x}[i], \mathbf{y}[i], \mathbf{p}[i]) + \text{Embed}_t(\Delta\mathbf{t})[i] \right),
\end{aligned} \quad (7)
$$

where $\boldsymbol{\rho}$, $\mathbf{x}$, $\mathbf{y}$, $\mathbf{p}$, and $\mathbf{t}$ are sequences representing the intensity factors, spatial coordinates, polarities, and timestamps of $L$ events. Here, $\Delta\mathbf{t}[0] = 0$ and $\Delta\mathbf{t}[i] = \tilde{\mathbf{t}}[i] - \tilde{\mathbf{t}}[i-1]$ for $i \in \{1, \dots, L-1\}$, where $\tilde{\mathbf{t}} = \mathbf{t}/\max(\mathbf{t})$. For a native event, $\boldsymbol{\rho}[i]$ is 1, while for a cluster event, it represents the number of raw events aggregated into that cluster. We take the logarithm of $\boldsymbol{\rho}$ to suppress clusters with an excessive number of events and prevent them from dominating the entire sequence.

To contextualize our formulation, it is instructive to compare event2vec with pioneering event-wise representations. Notably, EventNet (Sekikawa et al., 2019) also factorizes an event into a discrete spatial-polarity address and relative temporal information. Specifically, EventNet maps an event address $\mathbf{e}[i] = (\mathbf{x}[i], \mathbf{y}[i], \mathbf{p}[i])$ to a feature vector $\mathbf{z}[i] = h(\mathbf{e}[i])$, followed by a temporal coding function based on $\Delta t_{j,i} = \mathbf{t}[j] - \mathbf{t}[i]$. Since $\mathbf{e}[i]$ has only $2HW$ possible values, EventNet precomputes $h(\mathbf{e}[i])$ to implement it as a lookup table (LUT) during inference. In this sense, our standard spatial embedding in Eq. 2 shares a similar lookup

principle over the finite event-address space.

However, the two methods conceptualize and utilize time in fundamentally different ways. In EventNet, $\Delta t_{j,i}$ denotes the elapsed time between a past event and the latest event within a sliding window. This is processed by a hand-designed complex temporal coding function, which enables recursive event-wise updates through complex max aggregation within a PointNet-style backbone. In contrast, event2vec leverages the first-order interval $\Delta\mathbf{t}[i] = \tilde{\mathbf{t}}[i] - \tilde{\mathbf{t}}[i-1]$ between consecutive sampled or aggregated events. We feed this sequence of relative intervals into a learnable convolutional temporal embedding module. The resulting temporal representation is then additively fused with the spatial embedding to form Transformer-compatible event tokens, as formulated in Eq. 7.

In summary, while both EventNet and event2vec share the high-level principle of decoupling spatial-polarity from temporal information, their network architectures and computational targets diverge significantly. EventNet is tailored for asynchronous, recursive CPU processing using LUTs and symmetric max aggregation. Event2vec, conversely, is designed for GPU-efficient Transformer architectures utilizing learned embeddings. Furthermore, event2vec introduces a parametric spatial embedding to explicitly capture neighborhood semantics, along with an intensity factor $\boldsymbol{\rho}$ to encode aggregated event density—features that are conceptually absent in EventNet's formulation.

### 3.6. Network Structure

Figure 2 illustrates the network architecture, including (a) the spatial embedding architecture of event2vec, (b) the temporal embedding architecture of event2vec, and (c) the architecture of the entire network. Details can be found in Appendix A.2.

**Event2Vec:** The spatial embedding module $\phi$ consists of three linear layers. It gradually increases the number of features as $3 \rightarrow \frac{D}{4}$, $\frac{D}{4} \rightarrow \frac{D}{2}$ and $\frac{D}{2} \rightarrow D$. Layer Normalization (Ba et al., 2016) layers are also inserted after each linear layer to stabilize training. A ReLU activation is placed after the first two Layer Normalizations. The temporal embedding module has a similar structure to the spatial embedding module, except that it replaces linear layers with one-dimensional convolutional layers with a kernel size of 3 and a stride of 1. The first convolution maps the scalar interval sequence to $\frac{D}{4}$ channels, and the following two convolutions are depthwise convolutions. The number of channels gradually increases as $1 \rightarrow \frac{D}{4}$, $\frac{D}{4} \rightarrow \frac{D}{2}$ and $\frac{D}{2} \rightarrow D$.

**Backbone:** The backbone is constructed by stacking multiple Transformer blocks, each of which consists of a self-attention layer, a feed-forward network composed of two

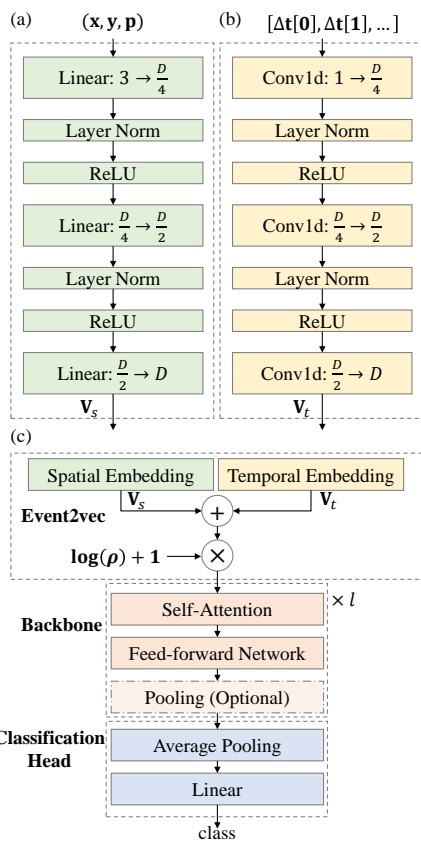

*Figure 2.* The network architecture for event classification using the event2vec representation.

fully connected layers, and an optional pooling layer to reduce the sequence length. We employ the Forgetting Transformer (Lin et al., 2025) as the self-attention in the backbone. Linear attentions such as Gated Linear Attention (Yang et al., 2024) can also be employed with a slight accuracy drop, the results of which are reported in Appendix A.3. It is important to recognize that the forgetting gates in Forgetting Transformer are order-sensitive. To enhance the learning capability, we extend the Forgetting Transformer to a parameter-shared bidirectional formulation. Further details are provided in Appendix A.4.

**Classification Head:** We employ an average pooling layer to aggregate features across all positions in the sequence. Then a linear layer is used to make a classification decision.

## 4. Experiments

We conduct a series of experiments on classification tasks using three neuromorphic datasets: DVS Gesture, ASL-DVS, and DVS-Lip. In this section, results are reported in the format $a \pm b$, representing the mean and standard deviation, respectively. For experiments that involve random sampling, results are computed over 10 independent runs on the test set.

### 4.1. Comparison Between Representations

**Accuracy and Parameter Efficiency** Table 1 compares the accuracy and model parameters of event2vec with those of other representations across the three datasets. Our models for DVS Gesture and ASL-DVS are trained directly on randomly sampled events. For DVS-Lip, our model first undergoes self-supervised pre-training (refer to Appendix A.5) on cluster events. We then report the fine-tuning accuracy on both randomly sampled events and cluster events generated by the proposed Batched K-Means++ algorithm. Our method achieves comparable accuracy on DVS Gesture and the highest accuracy on ASL-DVS and DVS-Lip among other leading representations, while demonstrating exceptional parameter efficiency. For example, previous state-of-the-art (SOTA) models use $2.79\times$, $815.93\times$, and $12.22\times$ as many parameters as our model on the three datasets.

**Throughput, Latency and Memory** We further compare model throughput and latency between our models and previous SOTA models, and results are shown in Table 2. The three models included for comparison in Table 2 are Max-Former (Fang et al., 2025), GNN & Transformer (Yuan et al., 2023), and Spiking ResNet18 & BiGRU (Dampfhoffer & Mesquida, 2024). All of these models provide official open-source code, which allowed us to conduct experiments based on their codebases. Appendix A.6 provides more details about these experiments. Throughput is a primary metric governing the efficiency of model training. For different models, we set the batch size to the largest possible power of 2 or the average of two adjacent powers of 2 (e.g., 64, 96, 128, ...) without exceeding the GPU memory limit, aiming to maximize the reported throughput. The results demonstrate that event2vec fully leverages the computational efficiency of Transformers, achieving training and inference throughput that is $4.21\times$ and $2.69\times$, $11.96\times$ and $62.67\times$, and $35.36\times$ and $5.70\times$ higher than those of prior works across the three datasets, respectively. For inference tasks on edge devices (e.g., an embedded neuromorphic system), the latency of processing a single event stream and the GPU memory consumed by the model are critical. We conducted comparative experiments and measured three latency components: event data preprocessing latency, model forward propagation latency, and total latency. The results show that across the three datasets, the total latency of our model is only $68.55\%$, $11.12\%$, and $14.68\%$ of that of the prior SOTA methods, while the memory consumption is merely $72.18\%$, $15.08\%$, and $68.35\%$.

### 4.2. Ablation Experiments

**Embedding Comparison** We conducted an ablation study on the DVS Gesture dataset to evaluate the accuracy contributions of different components, as detailed in Table 4. We tested various combinations of spatial embedding meth-

*Table 1.* Model performance and size comparison on neuromorphic datasets.

| Dataset | Method + Representation | Accuracy (%) | Params (MB) |
|---|---|---|---|
| DVS Gesture | Sparse GRU + Frame (Subramoney et al., 2023) | 97.80 | 4.80 |
| | SNN + Frame (Yao et al., 2023) | 98.23 | 6.50 |
| | FARSE-CNN + Window Slicing (Santambrogio et al., 2024) | 96.6 | 10.79 |
| | Event MAE + Point Cloud (Sun et al., 2025) | 97.75 | N/A |
| | Max-Former + Frame (Fang et al., 2025) | **98.6** | 1.45 |
| | Transformer + Event2Vec (4096 Random Events) | 97.57±1.31 | **0.52** |
| ASL-DVS | GNN, CNN + Graph (Bi et al., 2019) | 90.10 | 19.46 |
| | GNN & Transformer + Image & Voxel Graph (Yuan et al., 2023) | 99.60 | 220.30 |
| | Transformer + Event2Vec (1024 Random Events) | **99.91±0.05** | **0.27** |
| DVS-Lip | ResNet-18 & BiGRU + Frame (Tan et al., 2022) | 72.1 | 241.20 |
| | Spiking ResNet18 & BiGRU + Frame (Dampfhoffer & Mesquida, 2024) | 75.3 | 223.63 |
| | Transformer + Event2Vec (1024 Random Events) | 70.62±1.55 | **18.30** |
| | (1024 Batched K-Means++ Cluster Events) | **75.88** | |

*Table 2.* Throughput and latency comparison between Event2Vec and previous SOTA models.

| Dataset | Method | Batch Throughput (samples/s) | | Single-stream Inference | | | |
|---|---|---|---|---|---|---|---|
| | | Training | Inference | Latency (ms) | | | GPU Memory (MB) |
| | | | | Data Pre-processing | Forward | Total | |
| DVS Gesture | Max-Former | 241.12 ± 0.55 | 1077.35 ± 2.20 | 10.29 ± 0.23 | 23.89 ± 11.65 | 34.18 ± 11.75 | 834 |
| | Event2Vec | 1016.19 ± 61.18 | 2900.08 ± 277.30 | 9.92 ± 5.79 | 13.51 ± 9.04 | 23.43 ± 10.63 | 602 |
| ASL-DVS | GNN & Transformer | 78.08 ± 6.39 | 200.16 ± 12.66 | 55.12 ± 38.35 | 10.90 ± 0.45 | 66.02 ± 38.41 | 5464 |
| | Event2Vec | 933.58 ± 16.14 | 12543.81 ± 2565.54 | 1.10 ± 0.23 | 6.24 ± 2.89 | 7.34 ± 2.91 | 824 |
| DVS-Lip | Spiking ResNet18 & BiGRU | 10.85 ± 0.03 | 165.29 ± 2.25 | 373.82 ± 2.21 | 15.07 ± 7.73 | 388.89 ± 8.14 | 1226 |
| | Event2Vec | 383.71 ± 0.89 | 942.29 ± 22.36 | 17.23 ± 3.33 | 39.87 ± 1.28 | 57.10 ± 3.62 | 838 |

ods (standard (Eq. 2) vs. parametric (Eq. 5)) and temporal embedding modules (sinusoidal embedding on **t** vs. convolutional embedding on $\Delta$**t**). The combination of the standard embedding with our convolutional temporal embedding (Standard + Conv($\Delta$**t**)) yields the lowest accuracy. We attribute this to the standard embedding layer's lack of inductive bias, which prevents it from effectively learning neighborhood semantics and subsequently limits the performance of the convolutional temporal encoder. Consequently, when using our parametric embedding, the convolutional encoder achieves the highest accuracy. It is worth noting that our parametric embedding consistently outperforms the standard version when paired with any temporal embedding, validating the effectiveness of incorporating neighborhood semantics.

**Robustness to the Number of Events** Processing fewer events results in lower resource consumption, which is always desirable in event-based applications. Figure 3 compares our method with the sophisticated sampling techniques from (Araghi et al., 2025), which use a voxel grid representation. The results highlight the inherent effectiveness of event2vec: when paired with simple random sampling,

it consistently outperforms the voxel grid representation, even when the latter employs more complex, meticulously designed sampling strategies. Appendix A.7 provides additional details on the variations in the performance of the event2vec model with respect to the number of events.

**Robustness to Spatial Resolutions** We further tested the robustness of event2vec to variations in the spatial resolution of sensors and compared it with Max-Former (Fang et al., 2025) on the DVS Gesture dataset. For event2vec, the coordinates are treated as floating-point numbers, scaled to the target resolution $h \times w$ and then quantized. Subsequently, the coordinates are upscaled back to the original resolution $H \times W$ and re-quantized. For Max-Former, which adopts a frame-based representation, bilinear interpolation is used to downscale the frames to a lower resolution $h \times w$, followed by upscaling back to the original resolution $H \times W$. The results in Figure 4 demonstrate that event2vec exhibits strong robustness to changes in resolution. Moreover, it maintains a classification accuracy significantly higher than random guessing even when the resolution is reduced to $1 \times 1$ (i.e., complete loss of spatial information), which indirectly validates the effectiveness of the temporal embedding.

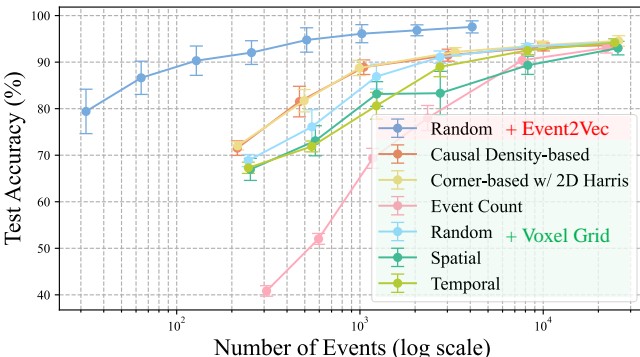

Figure 3. Accuracy vs. number of events compared to sampling techniques from (Araghi et al., 2025) on the DVS Gesture dataset.

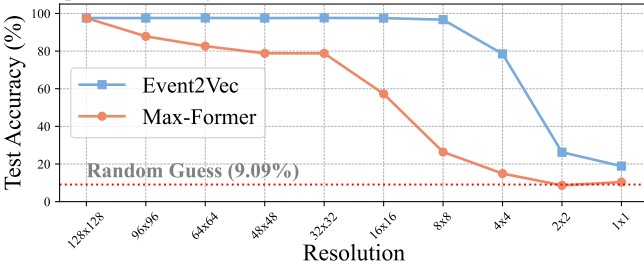

Figure 4. Accuracy vs. spatial resolution: Comparison with the SOTA Max-Former (Fang et al., 2025) on DVS Gesture.

**Evaluation of Representation Learning by Linear Probing** To verify whether the model can learn general feature representations, we adopted linear probing—a commonly used metric (Alain & Bengio, 2017; Radford et al., 2021)—for evaluation. Specifically, the event2vec model used for classifying DVS-Lip has the largest number of parameters among the classification models for the three datasets. Consequently, we utilized the model trained on DVS-Lip for linear probing on DVS Gesture and ASL-DVS.

We first examined whether DVS Gesture and ASL-DVS are inherently linearly separable. The results, presented in the first row of Table 3, show the linear probing accuracy without using event2vec or a Transformer. It can be observed that the accuracies for both datasets are significantly low. Next, we replaced the raw event inputs with embeddings encoded by event2vec; these results are shown in the second and third rows of Table 3. The Parametric Spatial + Convolutional Temporal embeddings proposed in this paper demonstrate a significant impact, with accuracy improvements exceeding 20 percentage points on both datasets. While the "Standard Spatial + Sin Temporal" baseline for event2vec performed worse than the parametric alternative, it still outperformed the raw events.

Furthermore, we utilized the event2vec component from the model trained on DVS-Lip, along with the first 5 layers of the 16-layer backbone network. We selected 5 layers because we found that this depth yields the optimal performance; using fewer or more layers would lead to a slight decline in performance. We froze the parameters of the

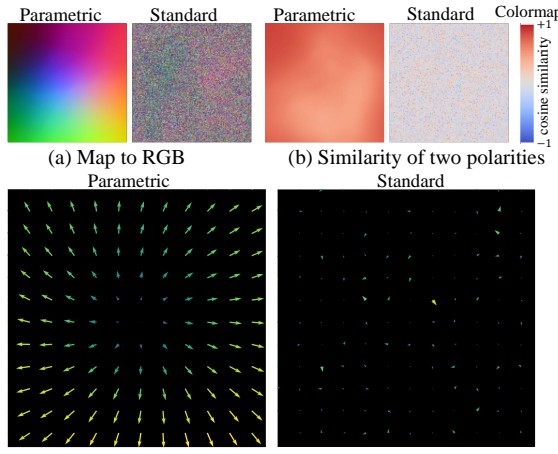

Figure 5. Visual comparison of the learned spatial embeddings.

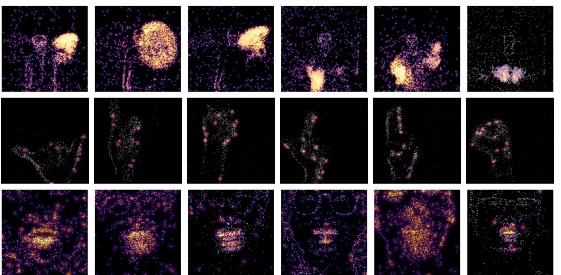

Figure 6. Event-level attention maps on samples from DVS Gesture (Row 1), ASL-DVS (Row 2), and DVS-Lip (Row 3).

extracted sub-model and appended a trainable classification head to it. The results are shown in the fourth and fifth rows of Table 3. As shown, the accuracy on both datasets continued to improve substantially. These results indicate that the model with event2vec and a multi-layer Transformer architecture can generate features with high linear separability when transferred from the training dataset to other datasets, demonstrating that the model has learned a general method for feature representation.

**Cluster Latency** Table 5 compares the average per-sample clustering latency on the test set, as well as the test-set accuracy of models trained with clustered data, when different K-Means clustering methods are applied to the DVS-Lip dataset. We compare our approach with two benchmarks: Scikit-learn's CPU-based K-Means (using K-Means++) and Meta's Faiss GPU K-Means (Johnson et al., 2019).As shown in Table 5, the proposed batched K-Means++ achieves the highest accuracy with latency comparable to the fastest setting. Notably, it is significantly faster than Scikit-learn and more accurate than the GPU-accelerated Faiss. While Faiss (iters=20) offers comparable speed, it suffers a 1.48% accuracy drop compared to our approach.

### 4.3. Visualization

**Neighborhood Semantics** To visually inspect the neighborhood semantics, we extract the spatial embedding weights

*Table 3.* Comparison of linear probing accuracy with different event2vec methods and with/without Transformer layers.

| Event2Vec | Transformer | DVS Gesture | ASL-DVS |
|---|---|---|---|
| None | None | 35.52±1.35% | 12.65±14.61% |
| Parametric Spatial + Convolutional Temporal | None | 56.32±2.56% | 38.90±10.48% |
| Standard Spatial + Sin Temporal | None | 37.36±2.22% | 26.63±10.91% |
| Parametric Spatial + Convolutional Temporal | First 5 layers | 86.94±1.21% | 69.83±7.66% |
| Standard Spatial + Sin Temporal | First 5 layers | 84.44±3.35% | 59.71±8.76% |

*Table 4.* Ablation analysis of embeddings on DVS Gesture.

| Spatial Embedding | Temporal Embedding | Accuracy (%) |
|---|---|---|
| Standard | Conv($\Delta$**t**) | 91.18±3.70 |
| Standard | Sin(**t**) | 93.16±2.19 |
| Parametric | Sin(**t**) | 96.56±1.46 |
| Parametric | Conv($\Delta$**t**) | 97.57±1.31 |

*Table 5.* K-Means clustering latency and accuracy on DVS-Lip.

| Method | Latency (ms) | Accuracy (%) |
|---|---|---|
| Batched K-Means++ (batch size = 64, iters=20) | 17.10±15.56 | 75.88 |
| Scikit-learn (iters=300) | 383.42 ± 149.22 | 75.08 |
| Scikit-learn (iters=20) | 374.22 ± 145.97 | 74.72 |
| Faiss (iters=300) | 162.41±126.24 | 74.12 |
| Faiss (iters=20) | 15.95±17.59 | 74.40 |

from models trained on the DVS Gesture dataset with the parametric ($\mathbf{W}_\phi$) and standard ($\mathbf{W}_s$) embedding layers. For each coordinate $(x, y, p)$, its $D$-dimensional embedding vector is projected onto a three-dimensional space using Principal Component Analysis (PCA). These 3D vectors are then interpreted as RGB color values and plotted at their corresponding $(x, y)$ locations to form an image. Figure 5(a) visualizes the resulting images for polarity 0 (images for polarity 1 are provided in Appendix A.8). The image derived from $\mathbf{W}_\phi$ displays smooth, continuous color gradients, akin to a color palette, indicating that spatially adjacent coordinates have semantically similar embeddings. In stark contrast, the image from $\mathbf{W}_s$ resembles random noise, signifying a lack of learned spatial correlation.

**Polarity Similarity** An object's edge moving across a pixel often triggers events of both polarities in close succession. We therefore hypothesize that the embeddings for opposite polarities at the same spatial location should also be semantically related. To test this, we compute the cosine similarity between the embedding vectors of the two polarities at each coordinate. As shown in Figure 5(b), the parametric embedding captures this relationship, exhibiting distinct regions of high similarity. Conversely, the similarity map for the standard embedding is predominantly close to zero, indicating that it fails to learn this inter-polarity correlation.

**Vector Field Representation** We visualize the learned spatial manifold as a vector field. The $D$-dimensional embedding vectors are projected onto their first two principal components using PCA. These resulting 2D vectors are then visualized using a quiver plot, where each arrow represents the direction and magnitude of the vector at its spatial coor-

dinate. Figure 5(c) illustrates the results. The vector field for the parametric embedding exhibits a coherent, laminar-like flow, revealing a smoothly structured semantic space. In contrast, the field for the standard embedding appears chaotic and turbulent, further confirming its inability to capture meaningful spatial relationships.

**Event-wise Attention** As event2vec is an event-wise representation, its attention mechanism can be visualized at a fine-grained, event-level resolution. Figure 6 displays attention heatmaps overlaid on the original event streams for DVS Gesture (row 1), ASL-DVS (row 2), and DVS-Lip (row 3). The visualizations reveal that the model correctly focuses on the hands in DVS Gesture, the finger joints and contours in ASL-DVS, and the lip region in DVS-Lip. However, consistent with the lower classification accuracy compared to the other two datasets, we also observe instances where the model incorrectly allocates significant attention to other facial features, such as the eyes and ears.

## 5. Conclusions

Neuromorphic event cameras introduce a paradigm shift in computer vision, presenting both unique opportunities and significant challenges. A central challenge has been reconciling their asynchronous, sparse nature with the synchronous, dense regular architectures of deep learning. In this paper, we introduced event2vec, a novel representation that directly addresses this challenge by enabling neural networks to natively process asynchronous events. Our experimental results demonstrate that event2vec achieves accuracy competitive with established methods while offering compelling advantages in parameter efficiency, preprocessing overhead, throughput, and robustness across varying numbers of events and spatial resolutions. The remarkable efficiency and robustness of event2vec suggest its significant potential for real-time deployment on resource-constrained edge devices, where low-latency sensing and low-power consumption are paramount. Beyond these performance metrics, the most significant contribution of event2vec is its conceptual alignment of event streams with the paradigm of natural language processing. This opens new avenues for research and application. By treating events as a sequential language, we can begin to explore novel applications by leveraging the sophisticated architectures developed for large language models.

## Acknowledgements

This work was supported in part by CoCoSys, a JUMP2.0 center sponsored by DARPA and SRC, the National Science Foundation (CAREER Award, Grant #2312366, Grant #2318152), the DARPA Young Faculty Award, the DoE MMICC center SEA-CROGS (Award #DE-SC0023198), and the Global Industrial Technology Cooperation Center (GITCC) program.

## Impact Statement

This paper presents work whose goal is to advance the field of Machine Learning. There are many potential societal consequences of our work, none of which we feel must be specifically highlighted here.

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

# A. Appendix

## A.1. Batched K-Means++ Cluster Algorithm

For challenging classification tasks like DVS-Lip, random sampling leads to significant information loss, resulting in suboptimal accuracy. To ensure that all events contribute to the final representation, we employ an event clustering approach. Given a raw event stream $\mathcal{E} = \{(x_i, y_i, t_i, p_i)\}_{i=0}^{N-1}$ containing $N$ events, our objective is to generate up to $L$ cluster events $\mathcal{R} = \{(x_{c,j}, y_{c,j}, t_{c,j}, p_{c,j}, \rho_j)\}$, where $(x_{c,j}, y_{c,j}, t_{c,j}, p_{c,j})$ denotes the coordinates of the $j$-th cluster center and $\rho_j$ represents the event count within that cluster. Notably, we perform clustering separately for the two polarities to prevent the loss of physical significance caused by polarity mixing. Specifically, assuming the number of events for each polarity is $N_0$ and $N_1$, we allocate clusters approximately in proportion to $N_0$ and $N_1$, while clipping the counts by the number of available events for each polarity. Finally, the two resulting cluster sets are merged and sorted chronologically.

The general practice of K-Means clustering involves using the K-Means function provided in Scikit-learn (sklearn) (Pedregosa et al., 2011), a Python machine learning library. However, this function is implemented on the CPU, resulting in slow execution when the number of events is large, which significantly increases the latency of the model in processing real-time tasks. To address this issue, we propose a GPU-based Batched K-Means++ Event Clustering algorithm, whose detailed workflow is presented in Algorithm 1. The acceleration of this algorithm mainly stems from the following optimizations:

1. **Reduced Loop Overhead**: The number of loop iterations is reduced by a factor of $B$.

2. **Parallel Computing**: Uses `torch.cdist` to compute distances from all points to the batch of $B$ new centers in parallel.

3. **Incremental Update**: The update of $\mathbf{D}^2$ only compares the current known minimum distance with the distance to the newly added batch of centers, avoiding recomputation against all previously selected centers.

## A.2. Model Structures and Hyper-parameters

Unless otherwise stated, all models were trained using BFloat16 mixed precision. The training configuration for all models includes a base learning rate of $lr_b = 0.001$, a per-GPU batch size of 64, and the AdamW optimizer (Loshchilov & Hutter, 2019) for 64 epochs. The effective learning rate is determined by a linear scaling rule based on the per-GPU batch size ($batch\_size$) and the number of GPUs ($n_{gpus}$) used in distributed data-parallel training: $lr = lr_b \cdot batch\_size \cdot n_{gpus}/256$. A warmup phase is implemented for the first 4 epochs, during which the learning rate is linearly increased from $0.01 \cdot lr$ to $lr$. For the subsequent epochs, a cosine annealing schedule (Loshchilov & Hutter, 2017) is employed to gradually reduce the learning rate to a minimum value, $lr_{min}$. For the DVS Gesture and ASL-DVS datasets, both weight decay and label smoothing were disabled. In contrast, for the DVS-Lip classification task, we set the weight decay to 0.05 and applied label smoothing with a factor of 0.1.

Table 6 provides a detailed summary of the model-specific hyperparameters. Here, $D$ denotes the embedding dimension, $l$ is the number of Transformer blocks in the backbone, $D_f$ represents the hidden feature dimension of the feed-forward neural network (FFN), and $n_{head}$ is the total number of attention heads. The `repeats` parameter specifies how many times the training set is iterated through within a single epoch. For Forgetting Transformer, the number of heads for the key ($\mathbf{k}$) and value ($\mathbf{v}$) projections is set to $\max(\lfloor n_{head}/2 \rfloor, 1)$, and RMS group normalization (Wu & He, 2018) is applied to the query and key projections. For the DVS Gesture and DVS-Lip experiments, gradient clipping is used to cap the $L_2$ norm of the gradients at 1.0.

For the DVS Gesture classification model, sequence average pooling with a stride of 2 is applied between backbone stages, whereas the other models do not use sequence pooling. The model for the DVS-Lip classification task was pre-trained on the DVS-Lip dataset using a self-supervised learning approach. This pre-training phase utilized a minimum learning rate of $lr_{min} = 10^{-6}$, a weight decay of 0.05, a `repeats` value of 3, and a masking ratio of 30%. Refer to Appendix A.5 for more details.

## A.3. Accuracy with Different Types of Self-Attention

In addition to the Forgetting Transformer (FoX), we also evaluated the performance when using Gated Linear Attention (GLA) (Yang et al., 2024), with the results presented in Table 7. The results show that GLA achieves high performance on

---

**Algorithm 1:** Batched K-Means++ Event Clustering on GPU

---

**Input:** Raw Event Stream $\mathcal{E} = \{(x_i, y_i, t_i, p_i)\}_{i=0}^{N-1}$, Total Points $N$
**Params:** Maximum Target Cluster Count $L$, Spatial Dimensions $H, W$, Batch Size $B$, Max Iterations $I_{max}$, Tolerance
      $tol$
**Output:** Cluster Event Set $\mathcal{R}$

/\* 1.   Data Preprocessing & Normalization                                                   \*/

1  Move the data to the GPU

2  Compute time span $t_{span} = t_{N-1} - t_0$ and normalized time $\hat{t}_i = (t_i - t_0)/t_{span}$

3  Construct 3D feature-space point set $\mathbf{V} = \{(x_i/(W-1), y_i/(H-1), \hat{t}_i)\}_{i=0}^{N-1}$

4  Split $\mathbf{V}$ into polarity-specific sets $\mathbf{V}^0$ and $\mathbf{V}^1$ based on polarity $\mathbf{p}$

5  Allocate polarity-specific target center counts approximately proportional to $N_0$ and $N_1$, clipped by the available events
    in each polarity

6  Initialize result set $\mathcal{R} = \emptyset$

/\* Cluster for each polarity separately                                           \*/

7  **for** *each point set* $\mathbf{V}_{sub} \in \{\mathbf{V}^0, \mathbf{V}^1\}$ *and target count* $K_{sub} \in \{L_0, L_1\}$ **do**

8      **if** $|\mathbf{V}_{sub}| == 0$ *or* $K_{sub} == 0$ **then**

9          **continue**

      /\* Phase 1:  Batched K-Means++ Initialization                   \*/

10      Randomly select the first center $\mathbf{c}_0 \in \mathbf{V}_{sub}$, initialize center set $\mathbf{C} = \{\mathbf{c}_0\}$

11      Compute squared distance from all points to first center $\mathbf{D}^2 = \|\mathbf{V}_{sub} - \mathbf{c}_0\|^2$

12      **while** $|\mathbf{C}| < K_{sub}$ **do**

13          Calculate sample count for this batch $M = \min(B, K_{sub} - |\mathbf{C}|)$

          /\* Parallel Sampling:  Use current distance as probability weights     \*/

14          Sample $M$ new candidate centers $\mathbf{C}_{new}$ based on weights $\mathbf{w} \propto \mathbf{D}^2$ without replacement

15          Add $\mathbf{C}_{new}$ to $\mathbf{C}$

          /\* Incremental Distance Update:  Only to newly added centers      \*/

16          $\mathbf{D}_{new}^2 = \min_{c \in \mathbf{C}_{new}} \|\mathbf{V}_{sub} - c\|^2$

17          Update global minimum distance $\mathbf{D}^2 \leftarrow \min(\mathbf{D}^2, \mathbf{D}_{new}^2)$

      /\* Phase 2:  Standard Lloyd's Iteration                       \*/

18      **for** $iter = 0$ ***to*** $I_{max} - 1$ **do**

19          **E-step:** Assign labels to points based on the nearest center in $\mathbf{C}$

20          **M-step:** Compute centroids of each cluster as the new $\mathbf{C}$

21          **if** *the center shift* $< tol$ **then**

22             **break**

      /\* Denormalization & Intensity Calculation                   \*/

23      Count points in each cluster as intensity $\boldsymbol{\rho}$

24      Map coordinates of $\mathbf{C}$ back to physical dimensions $(W-1, H-1, t_{span})$ and add the starting timestamp $t_0$

25      Add results $(x_{c,j}, y_{c,j}, t_{c,j}, p_{sub}, \rho_j)$ to $\mathcal{R}$

/\* 3.   Post-processing                                                          \*/

26  Merge all results in $\mathcal{R}$

27  Sort results by time $\mathbf{t}_c$ to restore chronological order

28  **Return** $\mathcal{R}$

---

*Table 6.* Hyper-parameters of training models for classification tasks on different datasets.

| Dataset | $D$ | $D_f$ | $n_{head}$ | $l$ | Repeats | $n_{gpus}$ | $lr_{min}$ |
|---|---|---|---|---|---|---|---|
| DVS Gesture | 64 | 128 | 2 | 4 | 24 | 4 | 0 |
| ASL-DVS | 64 | 128 | 2 | 2 | 1 | 7 | $10^{-6}$ |
| DVS-Lip | 192 | 384 | 6 | 16 | 3 | 4 | $10^{-6}$ |

*Table 7.* Comparison of accuracy with different types of self-attention.

| Dataset | Accuracy of FoX (%) | Accuracy of GLA (%) |
|---|---|---|
| DVS Gesture | $97.57 \pm 1.31$ | $96.67 \pm 0.67$ |
| ASL-DVS | $99.91 \pm 0.05$ | $99.85 \pm 0.12$ |
| DVS-Lip | 75.88 | 72.35 |

relatively easy classification tasks such as DVS Gesture and ASL-DVS; however, its performance drops by 3.53 percentage points on the more challenging DVS-Lip classification task, which may be attributed to the fact that linear attention struggles to prevent the decay of long-term memory when processing long input sequences. Overall, these results indicate that Event2Vec remains compatible with different attention mechanisms, while FoX is preferable for more challenging long-sequence tasks.

### A.4. Bidirectional Self-Attention

GLA is a typical linear attention mechanism, which can be regarded as a special case of recurrent neural networks (RNNs) (Katharopoulos et al., 2020), where the input sequence order affects the output results. Although FoX does not fall into the category of linear attention, it adopts an RNN-style gating mechanism that depends on input sequence order, and thus the input order also affects its outputs. Due to the fixed and limited size of hidden states, RNNs inevitably suffer from long-distance information attenuation when processing ultra-long sequences. To mitigate the degradation of long-term memory, we extend FoX to bidirectional variants.

We adapt this formulation to be bidirectional by inputting both forward and reversed $\mathbf{Q}, \mathbf{K}, \mathbf{V}$. The bidirectional outputs are computed as:

$$\mathbf{O}_f = \text{Attention}(\mathbf{Q}, \mathbf{K}, \mathbf{V}), \tag{8}$$

$$\mathbf{O}_b = \text{Reverse}(\text{Attention}(\overleftarrow{\mathbf{Q}}, \overleftarrow{\mathbf{K}}, \overleftarrow{\mathbf{V}})), \tag{9}$$

$$\mathbf{O}_{fb}[t] = \mathbf{W}_{fb}[\mathbf{O}_f[t]; \mathbf{O}_b[t]]. \tag{10}$$

Unlike classic bidirectional RNNs (Schuster & Paliwal, 1997) that often use independent parameters for each direction, our model employs shared parameters for the two directions. By sharing the projection weights ($\mathbf{W}_q, \mathbf{W}_k, \mathbf{W}_v, \mathbf{W}_{forget}$) across both passes, we ensure that the parameter count remains comparable to the unidirectional baseline, with the only increase arising from the fused output projection $\mathbf{W}_{fb}$.

We also tested the performance changes when using bidirectional self-attention without parameter sharing, and the results are summarized in Table 8. The results show that the number of parameters increases by approximately 25% when parameters are not shared. Although the fitting capacity is theoretically improved without parameter sharing, the test-set accuracy instead decreases, indicating slight overfitting. This experimental result demonstrates that our bidirectional self-attention with parameter sharing not only reduces the number of parameters but also mitigates overfitting.

*Table 8.* Changes in accuracy and parameters when using bidirectional self-attention without parameter sharing.

| Dataset | Parameters (MB) | Accuracy (%) |
|---|---|---|
| DVS Gesture | 0.65 (+25%) | 96.63 (-0.94) |
| ASL-DVS | 0.34 (+26%) | 99.86 (-0.05) |
| DVS-Lip | 22.90 (+25%) | 75.36 (-0.52) |

### A.5. Self-supervised Training Details

The event-wise nature of the event2vec representation lends itself well to self-supervised pre-training, which can significantly enhance model performance. Specifically, we adopt a masked modeling approach, akin to that used in BERT. The training objective is to mask the spatial coordinates and polarity $(x, y, p)$ of a subset of these events and train the model to predict the masked values based on the context provided by the surrounding events and their associated temporal information. This task compels the model to learn a meaningful understanding of spatiotemporal event patterns.

The self-supervised training framework is analogous to the Masked Language Model (MLM) objective in BERT (Devlin et al., 2019). Given a batch of spatial embedding tensors $\mathbf{v}_s$ of shape $(B, L, D)$, where $B$ is the batch size, $L$ is the sequence length, and $D$ is the embedding dimension, the process begins by randomly masking a portion of the input tokens.

A binary mask $\mathbf{m}$ of shape $(B, L)$ is generated by first sampling the starting positions of masked spans. For the DVS-Lip self-supervised configuration, the target mask ratio is $30\%$ and the span-length parameter is $l_{mask} = 10$; therefore, each token is selected as a span start with probability $0.3/10$. To prevent the model from making predictions via simple interpolation, each selected start position masks out a consecutive span of tokens. The length of each masked span is sampled as $\text{Geometric}(p) + 1$ with $p = 0.1$ and then truncated to a maximum length of 20. Each spatial token $\mathbf{v}_s[i][j]$ corresponding to a mask entry $\mathbf{m}[i][j] = 1$ is replaced by a single, learnable, $D$-dimensional mask token $\mathbf{v}_m$. This operation results in a corrupted spatial embedding tensor, which is then fused with the temporal embedding and intensity factor to form the corrupted event2vec tensor $\hat{\mathbf{v}}$. Concurrently, the original spatial coordinates and polarity $(\mathbf{x}_m, \mathbf{y}_m, \mathbf{p}_m)$ of the masked tokens are preserved to serve as the ground truth for the reconstruction loss.

The corrupted tensor $\hat{\mathbf{v}}$ is then processed by the model's FoX backbone. Following this, the output embeddings that correspond to the initially masked positions, denoted as $\hat{\mathbf{v}}_m$, are extracted from the final output tensor using the mask $\mathbf{m}$.

The objective is for the model to reconstruct the original spatial and polarity information from these corrupted embeddings. To achieve this, we first apply the inverse of the spatiotemporal fusion operation to isolate the spatial component of the reconstructed embeddings:

$$\hat{\mathbf{v}}_s = \frac{\hat{\mathbf{v}}_m}{\log(\boldsymbol{\rho}_m) + 1} - \mathbf{v}_{t,m}. \tag{11}$$

where $\boldsymbol{\rho}_m$ and $\mathbf{v}_{t,m}$ denote the intensity factors and temporal embeddings at the masked positions. The resulting tensor, $\hat{\mathbf{v}}_s$, is treated as the reconstructed spatial embedding. It is then passed through a decoder network, which mirrors the architecture of the spatial embedding encoder, to predict the original polarity and coordinates $(\hat{\mathbf{p}}, \hat{\mathbf{y}}, \hat{\mathbf{x}})$. Specifically, this decoder consists of a stack of linear layers, Layer Normalization, and ReLU activation functions. The network is designed to gradually reduce the feature dimension from $D$ down to 3. A tanh activation function is applied to the decoder output to constrain the predicted values to the range $(-1, 1)$. This aligns with the input preprocessing, where the ground-truth polarity and coordinates are also normalized to the same range.

Finally, the training objective is to minimize the mean squared error (MSE) loss between the predicted values $(\hat{\mathbf{p}}, \hat{\mathbf{y}}, \hat{\mathbf{x}})$ and the ground-truth values $(\mathbf{p}_m, \mathbf{y}_m, \mathbf{x}_m)$ of the masked tokens.

### A.6. Experimental Environment for Performance Testing

All experiments related to performance testing (such as throughput and latency) mentioned in this paper were conducted on a Red Hat Enterprise Linux 8.10 server. This server was equipped with an NVIDIA A100 GPU (80GB PCIe), an Intel Xeon Gold 6326 CPU (utilizing 8 cores), and 256GB of RAM. To mitigate the impact of data I/O, all datasets were loaded entirely into RAM for the duration of the experiments.

### A.7. Impact of the number of events on DVS Gesture

As illustrated in Figure 7, we benchmark the impact of varying the number of randomly sampled events ($L$) on several key metrics: training/inference throughput, single event stream inference latency, and accuracy on the DVS Gesture dataset. Experimental results on the ASL-DVS dataset show similar trends, and thus are not presented here. As $L$ increases, both training and inference throughput drop sharply, while the single-stream latency remains nearly unchanged. This indicates that for a single sample, the dominant latency arises from the CUDA kernel launch overhead rather than computation itself. Meanwhile, the accuracy improves rapidly with the growth of $L$, demonstrating that more events facilitate the model's decision-making process.

### A.8. Visualization of Neighborhood Semantics

Due to space constraints in the main paper, Figures 5(a) and 5(c) display visualizations for only a single event polarity. For completeness, this section provides supplementary visualizations that include both polarities. Figure 8 illustrates the embedding weights mapped to the RGB color space, while Figure 9 depicts them as a vector field.

### A.9. Data Augmentations

Transformers (including linear attention) lack inductive bias and thus require more data for learning. We used data augmentation methods to expand the data volume to a certain extent, thereby improving performance. Specifically, we

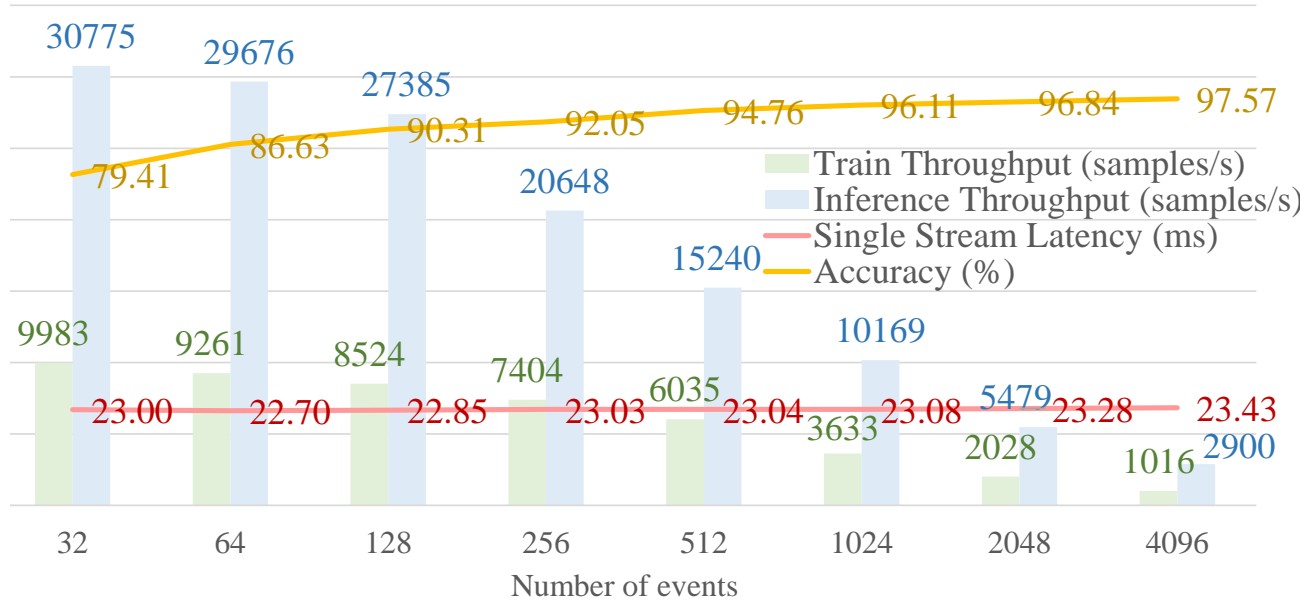

*Figure 7.* Effect of number of events on DVS Gesture.

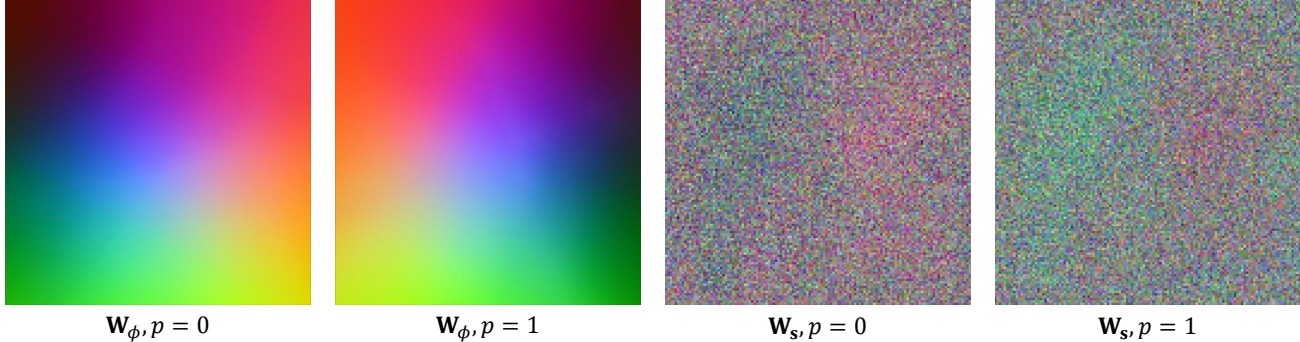

*Figure 8.* Visualization of the parametric embedding weight $\mathbf{W}_\phi$ and the standard embedding weight $\mathbf{W}_s$ in the RGB domain.

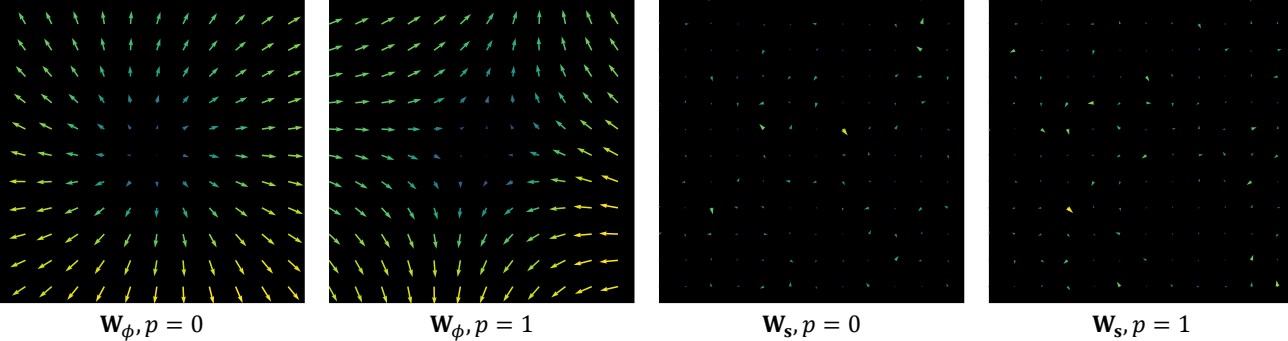

*Figure 9.* Visualization of the parametric embedding weights $\mathbf{W}_\phi$ and the standard embedding weight $\mathbf{W}_s$ as a vector field.

did not use data augmentation on the ASL-DVS dataset because we found that state-of-the-art (SOTA) performance could be achieved without it; this is likely due to the sufficient scale of this dataset: the number of samples in its training set is approximately 80,640, while that of DVS Gesture is 1,176, and DVS-Lip is 14,896.

Denote $\mathcal{U}(a, b)$ as the uniform distribution between $a$ and $b$, and $\mathrm{RandInt}(m, n)$ as a random integer taken from the set $\{m, m + 1, \ldots, n\}$, where each integer has an equal probability of being selected.

For an event stream, the data augmentations are applied to events directly. For simplicity, we omit the event index in this subsection. When a stochastic transform is enabled, its random parameters are sampled for each event stream in the batch unless otherwise specified. Note that the coordinates are converted to floating-point precision before applying any augmentation. After all augmentations are applied, only events whose coordinates are valid, i.e., $x \in [0, W - 1], y \in [0, H - 1]$, are kept.

For the classification task on DVS Gesture, the random-apply policy independently enables each of the following transformations with a probability of 0.6:

- **Random Resizing**: Coordinates $(x, y)$ are scaled to $(s_x \cdot x, s_y \cdot y)$, with scaling factors $s_x, s_y \sim \mathcal{U}(0.8, 1.2)$.

- **Random Rotation**: Coordinates are rotated by an angle $r \sim \mathcal{U}(-10, 10)$ degrees.

- **Random Shearing**: A shear transformation is applied with factors $\lambda_x, \lambda_y \sim \mathcal{U}(-0.02, 0.02)$.

- **Random Translation**: Coordinates are translated by offsets $d_x, d_y \sim \mathcal{U}(-16, 16)$.

- **Random Erasing**: An $h \times w$ area with $h, w \sim \mathcal{U}(0, 16)$ is erased with a probability of 0.1. The center of this area $(c_x, c_y)$ satisfies $c_x \sim \mathcal{U}(0, W - 1), c_y \sim \mathcal{U}(0, H - 1)$.

- **Temporal Chunk Dropout**: Eight candidate temporal chunks are removed from the event stream. Let $L_{valid}$ be the number of valid events. The length of each removed chunk is sampled as $l_{chunk} = \mathrm{RandInt}(1, 256) \cdot L_{valid}/L$, and its starting position is sampled uniformly from the token indices.

During the self-supervised phase of the model for classifying DVS-Lip, a series of geometric transformations are employed. The random-apply policy independently enables each listed transform with a probability of 0.5:

- **Random Resizing**: Coordinates $(x, y)$ are scaled to $(s_x \cdot x, s_y \cdot y)$, with scaling factors $s_x, s_y \sim \mathcal{U}(0.8, 1.2)$.

- **Random Rotation**: Coordinates are rotated by an angle $r \sim \mathcal{U}(-15, 15)$ degrees.

- **Random Shearing**: A shear transformation is applied with factors $\lambda_x, \lambda_y \sim \mathcal{U}(-0.05, 0.05)$.

- **Horizontal Flipping**: The event stream is flipped horizontally with an internal probability of 0.5.

- **Random Translation**: Coordinates are translated by offsets $d_x, d_y \sim \mathcal{U}(-16, 16)$.

When training the model for classifying DVS-Lip, we use the following data augmentations:

- **Random Resizing**: Coordinates $(x, y)$ are scaled to $(s_x \cdot x, s_y \cdot y)$, with scaling factors $s_x, s_y \sim \mathcal{U}(0.8, 1.2)$.

- **Random Rotation**: Coordinates are rotated by an angle $r \sim \mathcal{U}(-15, 15)$ degrees.

- **Random Shearing**: A shear transformation is applied to $x$ and $y$ with shear factors $\lambda_x, \lambda_y \sim \mathcal{U}(-0.05, 0.05)$.

- **Random Flip**: The event stream is flipped horizontally with a probability of 1.

- **Random Translation**: Coordinates $x$ and $y$ are translated by offsets $d_x, d_y \sim \mathcal{U}(-16, 16)$.

- **Random Erasing**: An $h \times w$ area with $h, w \sim \mathcal{U}(0, 16)$ is erased with a probability of 0.1. The center of this area $(c_x, c_y)$ satisfies $c_x \sim \mathcal{U}(0, W - 1), c_y \sim \mathcal{U}(0, H - 1)$.

*Table 9.* Data augmentation strategies for methods in Table 1.

| Dataset | Method + Representation | Data Augmentations |
|---|---|---|
| DVS Gesture | Sparse GRU + Frame (Subramoney et al., 2023) | Random crop, translation, and rotation |
| | SNN + Frame (Yao et al., 2023) | Random slice and integrate |
| | FARSE-CNN + Window Slicing (Santambrogio et al., 2024) | Random coordinate translations |
| | Event MAE + Point Cloud (Sun et al., 2025) | Point resampling from Point-BERT (Yu et al., 2022) |
| | Max-Former + Frame (Fang et al., 2025) | Mixup and Cutmix |
| | Transformer + Event2Vec | Random resizing, rotation, shearing, translation, erasing, and chunk dropout |
| ASL-DVS | GNN, CNN + Graph (Bi et al., 2019) | Random scale, flip, and rotation of node positions |
| | GNN & Transformer + Image & Voxel Graph (Yuan et al., 2023) | Random scale and translate |
| | Transformer + Event2Vec | None |
| DVS-Lip | ResNet-18 & BiGRU + Frame (Tan et al., 2022) | Random crop and horizontal flip |
| | Spiking ResNet18 & BiGRU + Frame (Dampfhoffer & Mesquida, 2024) | Random crop, horizontal flip, spatial masking, zoom, and temporal mask |
| | Transformer + Event2Vec | Random resizing, rotation, shearing, flipping, translation, erasing, and chunk dropout |

- **Random Chunk Drop**: Four candidate temporal chunks are removed from the event stream. Let $L_{valid}$ be the number of valid events. The length of each removed chunk is sampled as $l_{chunk} = \text{RandInt}(1, 128) \cdot L_{valid}/L$, and its starting position is sampled uniformly from the token indices.

The random-apply policy independently enables each augmentation listed above with a probability of $0.5$. TokenMix is applied to the embedding tensor with a probability of $0.5$. Specifically, when training on cluster events, the intensity $\rho$ is randomly set to $1$ with a probability of $0.1$. We use drop path (Larsson et al., 2016) in the FoX backbone, with the probability increasing linearly from $0$ to $0.4$ with depth.

In addition, all other methods in Tab. 1 also use data augmentation, which is summarized in Tab. 9.

