# OpenReview forum: "Event2Vec: Processing neuromorphic events directly by representations in vector space"
_ICML.cc/2026/Conference — ICML 2026 regular_

### Official Review · Reviewer_3KtX · 2026-03-06

**Soundness:** 3
**Presentation:** 3
**Significance:** 3
**Originality:** 3
**Overall Recommendation:** 5
**Confidence:** 4

**Summary:**

This paper introduces event2vec, a novel representation that directly addresses this challenge by enabling neural networks to natively process asynchronous events. Experimental results demonstrate that event2vec achieves accuracy competitive with established methods while offering compelling advantages in parameter efficiency, pre-processing overhead, throughput, and robustness across varying numbers of events and spatial resolutions.

**Compliance With Llm Reviewing Policy:**

Affirmed.

**Key Questions For Authors:**

See weaknesses.

**Limitations:**

Yes.

**Strengths And Weaknesses:**

Strengths

1. By linking event encoding with word embeddings in NLP, this approach solves the issue of current encoding methods losing the asynchronous and sparse nature of the event data. It provides an excellent reference for downstream tasks using event cameras.

2. The authors evaluated their method on three datasets (DVS Gesture, ASL-DVS, and DVS-Lip) with excellent results. They also included plenty of ablation studies and visualizations to validate their design.

3. The paper is well-written and clearly expressed.

Weaknesses

1. The paper's layout could be improved. The abstract is too long. Additionally, the introduction typically ends after the contributions are listed, but the authors added extra text to emphasize their contributions again, which feels redundant.

2. The method section does not include figures that specifically illustrate their proposed approach, making it hard to intuitively understand the design.

---

> ### Author Rebuttal · Authors · 2026-03-30
>
> ## Paper Layout and Redundancy
> We also acknowledge that the abstract is overly long. We will condense it in the revision by removing expository sentences that overlap with the Introduction, while retaining the core problem statement, method description, and key results.
>
> We have re‑examined the main text and agree that the content in Lines 82–97 on the right is indeed redundant. We will remove it in the revised version.
>
> ## Architectural Illustrations
> We sincerely thank the reviewer for this constructive feedback. We completely agree that a clear architectural diagram is essential for readers to intuitively grasp the Event2Vec design.
> We would like to respectfully direct your attention to Figure 6 in Appendix A.2 (Page 15). This figure provides a comprehensive, step-by-step visual illustration of our proposed network architecture, explicitly detailing the Spatial Embedding, Temporal Embedding, the fusion process, and the Transformer backbone.
>
> During the initial submission, strict page constraints unfortunately forced us to relegate this core figure to the supplementary material. Following your highly valuable suggestion, we will absolutely incorporate Figure 6 (or a slightly streamlined version of it) directly into the main Method section of the revised manuscript. We believe this addition will significantly enhance the readability and intuitiveness of our proposed approach for all future readers.

---

> > ### Author Rebuttal · Reviewer_3KtX · 2026-04-01
> >
> > My concerns have been adequately addressed.

---

### Official Review · Reviewer_wC7c · 2026-03-10

**Soundness:** 4
**Presentation:** 4
**Significance:** 3
**Originality:** 3
**Overall Recommendation:** 5
**Confidence:** 4

**Summary:**

This paper proposes Event2Vec, a representation framework that embeds neuromorphic events into a vector space so that event streams can be processed directly by Transformer-based models. Each event is represented using a spatial embedding derived from (x,y,p) and a temporal embedding learned from timestamp differences. A parametric spatial embedding introduces spatial inductive bias, while temporal information is modeled through convolution over timestamp differences. Experiments on three benchmarks (DVS Gesture, ASL-DVS, and DVS-Lip) demonstrate competitive accuracy with strong parameter efficiency and throughput.

**Compliance With Llm Reviewing Policy:**

Affirmed.

**Final Justification:**

The author has resolved most of my questions; I will maintain my rating.

**Key Questions For Authors:**

1.Can this event representation be applied to more complex tasks, such as object detection? In addition, can it meet the real-time requirements in scenarios such as autonomous driving?

2.Is the modeling of time differences reasonable? Two events that are adjacent in time may have no spatial relation. For example, they may simply reflect motions from different parts of the scene, such as events generated by the movement of a finger and an arm. Although these events are not spatially adjacent, they may still appear consecutively in time.

**Limitations:**

See questions and weaknesses

**Strengths And Weaknesses:**

Strengths:

1.The paper presents a relatively novel perspective by drawing an analogy between neuromorphic events and tokens in natural language processing, and proposes an encoding scheme based on this ide

2.The experimental results are strong, showing that the method can achieve competitive performance even when using a very small num

3.The paper is clearly written, with a well-structured and logically organized presentation of the method and experiments.

Weaknesses:

1.The experiments in this paper are conducted only on classification tasks. Demonstrating its performance on other downstream tasks would further strengthen the paper’s claims.

2.The main contribution of this work lies in the representation design, while the network architecture itself shows limited novelty and largely relies on existing components.

---

> ### Author Rebuttal · Authors · 2026-03-30
>
> ## Evaluation on Downstream Tasks and Real-time Requirements
>
> We completely agree that demonstrating Event2Vec on dense downstream tasks is an important next step. The primary goal of this manuscript is to introduce a fundamentally novel event representation paradigm. We deliberately focused on three widely used classification benchmarks to rigorously establish its advantages in parameter efficiency, throughput, and robustness as a foundational contribution.
>
> While full downstream evaluations are left for future work, our manuscript already contains empirical evidence of regression capability. Object detection fundamentally relies on dense coordinate regression. As detailed in Appendix A.5, our self-supervised Masked Modeling pre-training requires the network to explicitly regress the exact continuous spatial coordinates $(x, y, p)$ of masked events via MSE loss, decoded through a network mirroring the spatial embedding encoder to map vectors back to continuous coordinates. This directly demonstrates that Event2Vec preserves the fine-grained spatial topology necessary for dense regression tasks. Establishing this robust foundational representation was our first priority, and extending Event2Vec to complex downstream tasks is the immediate focus of our ongoing research.
>
> Regarding real-time requirements: Table 2 shows that Event2Vec achieves a total single-stream inference latency of just 57.10 ms on DVS-Lip, compared to 388.89 ms for the SOTA Spiking ResNet18 & BiGRU—a nearly 7× speedup. Furthermore, it entirely eliminates the pre-processing overhead of graph- or frame-based methods. Because the representation is inherently lightweight and highly optimized for parallel hardware, we are confident this computational efficiency will translate to real-time object detection frameworks. Extending Event2Vec to event-based object detection for autonomous driving is the immediate focus of our ongoing research.
>
> ## Novelty of Network Architecture
>
> We agree that our network architecture relies on existing components. However, this architectural simplicity is a deliberate design choice and one of our most significant contributions. Historically, processing neuromorphic events has forced a difficult dilemma: researchers must either convert events into dense synchronous frames (which destroys their inherent sparsity and high temporal resolution) or adopt specialized irregular architectures such as Spiking Neural Networks (SNNs), Graph Neural Networks (GNNs), or Sparse CNNs. Unfortunately, these irregular architectures consistently fail to fully exploit the massive parallel computing capabilities of modern GPUs.
>
> Event2Vec resolves this bottleneck. By representing sparse, asynchronous events in a continuous vector space, we decouple the neuromorphic data modality from the network architecture, enabling direct processing by a standard Transformer backbone. In other words, we no longer need to reinvent complex, task-specific network structures for event data. The fact that Event2Vec achieves competitive performance using an "unoriginal" standard Transformer is precisely the point: it proves the exceptional generality, hardware efficiency, and plug-and-play compatibility of our representation.
>
> ## Rationale for Temporal Difference Modeling
>
> You are correct that consecutive events in the 1D sequence rarely share a spatial origin—the finger-and-arm example is apt. However, our architecture explicitly resolves this spatial discontinuity and captures holistic, coordinated actions through a two-step process:
>
> - **Capturing Concurrency via Decoupled Temporal Density ($\Delta t$):** Spatial and temporal embeddings are strictly decoupled. The 1D convolution operates exclusively on $\Delta t$, entirely independent of spatial coordinates. Its purpose is not to enforce spatial smoothness, but to act as a temporal preconditioner capturing local event density or "action velocity". When distant body parts move simultaneously, their interleaved events produce extremely small $\Delta t$ values, providing a strong signal about the intensity and pacing of the concurrent movement.
>
> - **Global Routing via Self-Attention:** Because our Parametric Spatial Embedding explicitly preserves 2D coordinates, the network never conflates the finger with the arm. Once in the Transformer, self-attention computes pairwise similarities across the full sequence, dynamically routing and associating spatially distant but temporally concurrent events—precisely analogous to how attention effortlessly bridges semantically discontinuous adjacent words in NLP to build coherent global context.
>
> **Empirical Validation:** This capability is visually confirmed in Figure 5 (Row 2): on ASL-DVS, event-level attention maps show the model correctly focusing on spatially distributed but functionally correlated keypoints (finger joints and hand contours) to comprehend the complete gesture.

---

> > ### Author Rebuttal · Reviewer_wC7c · 2026-04-02
> >
> > The author has resolved most of my questions; I will maintain my rating.

---

### Official Review · Reviewer_jKHr · 2026-03-10

**Soundness:** 3
**Presentation:** 4
**Significance:** 4
**Originality:** 4
**Overall Recommendation:** 5
**Confidence:** 5

**Summary:**

This paper addresses the representation problem of event camera data and proposes a method called Event2Vec. The core idea is to directly encode raw events into vector sequences that can be fed into Transformers, rather than first converting the event stream into dense intermediate representations such as event frames or voxel grids. This approach better preserves the inherent sparsity, asynchronicity, and temporal information of event data. In terms of methodology, the paper constructs separate spatial and temporal event embeddings, organizes them into a unified representation, and then combines this with Transformers to accomplish downstream classification tasks. Overall, this work falls into the category of "direct representation and modeling for event data." The experimental section primarily validates the approach on multiple event classification benchmarks, supplemented by ablation studies and efficiency comparisons, demonstrating that this representation method achieves a favorable balance between accuracy and computational efficiency.

**Compliance With Llm Reviewing Policy:**

Affirmed.

**Final Justification:**

After the rebattle, I am confident in the evaluation and review comments I provided. Most of my concerns have been addressed, and I believe this paper should be accepted because it is genuinely important.

**Key Questions For Authors:**

1. In tasks requiring precise coordinate output, would Event2Vec fail? Consider an object localization task where the network output should be explicit bounding box coordinates. When using Event2Vec to process event data, the positional information in the network input is encoded in 1D features. Would this cause a mismatch between the input spatial positions (1D) and output spatial positions (2D), making it completely impossible for the network to learn this task? This concern is especially relevant in mainstream detection paradigms where convolutions extract object edge information. In other words, can Event2Vec be properly applied to regression tasks, and do the authors have any insights on this issue that were not written in the paper?

**Limitations:**

Yes

**Strengths And Weaknesses:**

Strengths:

1. Very detailed method description, presentation, and visualization, with comprehensive and fair experiments.

2. Existing event aggregation methods have consistently faced a problem: only frame-based methods can achieve truly competitive performance, but at the cost of completely abandoning asynchronous characteristics; asynchronous algorithms like graph neural networks perform poorly and cannot scale to very large sizes. This paper genuinely solves this problem by proposing a method that is both asynchronous and capable of leveraging cutting-edge neural network research. I believe this may be the most important research in the field of event representation and even event camera research in recent years. I am quite confident in this assessment.

Weaknesses:

1. This work appears somewhat exploratory due to the lack of evaluation on other classic vision tasks (optical flow estimation, object detection, image restoration and denoising, etc.). However, this allows for more detailed descriptions and insights that readers would otherwise struggle to grasp without the authors' explanation. Therefore, I do not consider this a detracting factor.

2. The authors suggest that this work has the potential to push event cameras toward large language model research, but no experiments in this direction were actually conducted, even though it is theoretically feasible. From another perspective, this work was only evaluated on three relatively small datasets with simple tasks and rich spatiotemporal information. Experiments on datasets like N-ImageNet might be more convincing—while they are synthetically generated, they are captured by real event cameras, so I believe they possess genuine spatiotemporal characteristics.

---

> ### Author Rebuttal · Authors · 2026-03-30
>
> ## Scope of Evaluated Tasks
>
> We agree that extending Event2Vec to dense prediction and low-level vision tasks is the crucial next step. Because processing asynchronous events directly in a vector space is a novel paradigm, this paper deliberately focuses on establishing a solid conceptual and empirical foundation. Extending to tasks such as optical flow estimation, object detection, and image restoration is essential for achieving industrial-grade applicability and demonstrating superiority over frame-based representations. We are actively exploring these directions as the primary focus of our ongoing research. We sincerely thank you for recognizing this focused scope is not a detracting factor. We are glad our detailed explanations successfully conveyed Event2Vec's intuition, balancing the lack of downstream tasks.
>
> ## LLM Potential and N-ImageNet
>
> 1. Regarding the potential for LLM research: We completely agree that scaling this up to a true "Event LLM" is an exciting direction. However, the primary scope of this paper is foundational representation learning. By introducing Event2Vec, our goal is to build the essential "infrastructure" that allows event streams to be natively and efficiently processed by standard Transformer architectures. Training a generalized Event LLM requires massive, diverse pre-training datasets and computational resources, which extends well beyond the scope of introducing a new representation paradigm. Nevertheless, we have taken the first step: as detailed in Appendix A.5, we successfully implemented a BERT-style self-supervised Masked Modeling pre-training on the Event2Vec representation. This serves as a strong proof-of-concept that our representation is fully compatible with LLM-style training methodologies.
>
> 2. Regarding the evaluation on N-ImageNet vs. DVS-Lip:
> We respectfully disagree that N-ImageNet possesses "genuine spatiotemporal characteristics" representative of complex real-world dynamics. N-ImageNet is constructed by recording a monitor displaying moving 2D static images, so the generated events stem purely from global uniform motion (ego-motion), entirely lacking independent non-rigid object motion, occlusions, and depth variations.
> In contrast, DVS-Lip captures highly complex, non-rigid, localized dynamic deformations (e.g., rapid and subtle mouth movements), representing a far more rigorous stress-test of fine-grained spatiotemporal semantics than global translations of static images. We therefore argue that DVS-Lip is a more faithful and challenging proxy for evaluating real-world neuromorphic vision models.
> We fully agree that evaluating Event2Vec on even larger-scale, truly dynamic real-world datasets is a vital next step. Unfortunately, scaling up the training on such massive datasets from scratch requires computational resources and training cycles that exceed the limited time window of the rebuttal period.
>
> ## **Applicability to Regression Tasks**
>
> We completely understand your intuition, but we would like to clarify that Event2Vec is fully capable of dense coordinate predictions; there is no inherent mismatch between the 1D input sequence and 2D spatial outputs, for the following reasons.
>
> 1. **Empirical Evidence (Appendix A.5):** A coordinate regression task is already included and successfully executed in our manuscript. The self-supervised pre-training for DVS-Lip trains the model via a Masked Modeling objective to predict the exact spatial coordinates of masked events, minimizing MSE between predicted and ground-truth coordinates. The output is decoded through a network mirroring the spatial embedding encoder (Figure 6( c )), mapping vectors directly back to continuous spatial coordinates. This empirically proves that Event2Vec embeddings retain precise, decodable positional information.
>
> 2. **Clarifying the 1D Sequence Space vs. the 2D Feature Space:** These two spaces are fundamentally orthogonal: the sequence index carries no spatial meaning. The 1D sequence structure (length $L$) is simply a formatting requirement for the Transformer. The true 2D spatial topology ($x$, $y$ coordinates) is not flattened or lost; it is completely preserved and losslessly encoded into the $D$-dimensional continuous feature vector via our Parametric Spatial Embedding. As shown in Figure 4( c ), this encoding constructs a smooth spatial manifold that preserves neighborhood semantics, confirming that 2D spatial structure survives the 1D formatting.
>
> 3. **Global Reconstruction via Self-Attention (The DETR Analogy):** A perfect and proven analogy in mainstream vision is DETR (DEtection TRansformer), which flattens 2D image features into a 1D sequence yet successfully outputs precise 2D bounding box coordinates for object localization. Event2Vec operates on the same mathematical foundation: a 1D sequence of tokens enriched with 2D spatial embeddings, from which Self-Attention reconstructs the physical 2D topology by computing pairwise token similarities.

---

> > ### Author Rebuttal · Reviewer_jKHr · 2026-04-02
> >
> > The author’s response addressed my concerns very well, and I will keep my score unchanged.

---

### Official Review · Reviewer_KJex · 2026-03-18

**Soundness:** 3
**Presentation:** 4
**Significance:** 4
**Originality:** 3
**Overall Recommendation:** 3
**Confidence:** 5

**Summary:**

Inspired by word2vec, the paper presents a method for mapping individual events to a vector space using a combination of spatial and temporal embedding functions. These functions are well-suited to pretraining with an objective inspired by masked language models. The paper then discusses two strategies to reduce the resulting vectors, either by a batched implementation of K-means++ or by random subsampling.  When combined with a Forget Transformer, it is shown that the resulting vectors improve classification scores over related work, both as a function of the number of inserted events and spatial resolution. The resulting models require orders of magnitude fewer parameters, less memory, and exhibit higher throughput and during training.

**Compliance With Llm Reviewing Policy:**

Affirmed.

**Key Questions For Authors:**

* How does linear probing compare when applied to raw events, and how would the transformer fare if presented with raw events?
* How would the proposed method perform clustering on sequences of events that are presented one at a time, instead of one sequence at a time?
* What is the reason for showing different methods for different datasets?

**Limitations:**

Yes

**Strengths And Weaknesses:**

**Strengths**
The paper is well written, and the motivation for converting events into higher-dimensional embeddings is strong. The results show that the method is effective at getting high classification accuracies with only a few events, which shows how informative they are. The paper presents qualitative results showing the spatial regularity of the embedding space and that the attention of events is correctly centered around semantically meaningful parts of the stream, underlining the effectiveness of the embeddings.

**Weaknesses**
* While the results show the effectiveness of the embeddings, I do not fully understand the motivation of using K-means++ clustering to reduce the event stream, since this presumably needs to be done on a per-stream basis, and may therefore introduce stability issues if this were done on an ongoing stream. A discussion of how the authors envision the clustering to be done in a streaming fashion would be much appreciated.
* The embedding modules mirror the ones in Sekikawa CVPR'19 closely, in that they similarly separate spatial and temporal embedding, but use a point-net backbone. Thus, a clear mention of their method and the similarity with Eq. 2 would be in order.
* In the method section, I believe that the 1D convolution over Delta t's must suffer from some spatial discontinuities, since subsequent events in the list might not necessarily come from the same pixel. Is there a misunderstanding here, or how is this discontinuity handled in practice?
* The linear probing experiment was only conducted on the proposed representation, and it is thus not clear if a similar result could not have been achieved by simply training on the raw events. Similarly, a comparison of Linear Attention + raw events would illuminate the importance of the representation, as opposed to the network backbone.
* It is not clear to me why different methods are compared for different datasets in Tabs. 1-2. I would also appreciate citations for the various methods, and a brief discussion of their main difference to the proposed approach.

**Minor**
* Eq. 5 should have something like o(Delta x, Delta y)

---

> ### Author Rebuttal · Authors · 2026-03-30
>
> ## Online Clustering
> We agree that streaming processing is highly desirable. However, this limitation is shared by most frame-based baselines, which also require the full sequence upfront (partitioning by the last event's timestamp or index) rather than operating in a true streaming fashion.
> Nevertheless, Event2Vec can immediately support strict one-at-a-time streaming via Reservoir Sampling. For a target length $L$, we maintain an array of $L$ events. When the $i$-th event arrives ($i > L$), it replaces an existing event at a random index $j \in [1, i]$ (if $j \le L$), or is otherwise discarded. As shown in Table 1, this streaming-compatible ramdom sampling approach already achieves a competitive $70.62\%$ accuracy on the challenging DVS-Lip dataset, demonstrating practical effectiveness.
>
> ## Similarity to Sekikawa CVPR'19
> We cited and briefly introduced Sekikawa et al. (2019) in our current version. During inference, they process the MLP via a Look-Up Table (LUT), which is fundamentally identical to Eq. 2. They also separately encode the spatial coordinates $(x, y, p)$ and timestamp differences $\Delta t$, matching our decomposition. We will explicitly note these similarities in Section 3 of the next revision.
>
> ## Discontinuity of $\Delta t$
> Due to word count limitations, please refer to the response "Rationale for Temporal Difference Modeling" to reviewer wC7c.
>
> ## Linear Probing
> Since Transformers require $[L, D]$ inputs, raw events $[L, 4]$ need a projection; we use Standard Spatial + Sinusoidal Temporal Embeddings as the bias-free baseline. We expanded our linear probing experiments across three configurations: **1)** pure raw events $[x, y, t, p]$, **2)** representation only (no Transformer), and **3)** representation + first 5 Transformer layers.
>
> **Key takeaways:**
>
> - **Raw events are entangled:** Direct linear probing on raw events yields very low accuracy (see R1).
> - **Event2Vec is effective standalone:** Even without the Transformer, linear probing on Event2Vec achieves **56.32%** on DVS Gesture, far exceeding pure raw events (**35.52%**) and the baseline embedding (**37.36%**). Our embeddings actively disentangle features.
> - **Consistent superiority:** Parametric Spatial + Convolutional Temporal embeddings consistently outperform the baseline embeddings, with or without the backbone.
>
> R1：The accuracy of linear regression on the original events
>
> | Dataset      | DVS-Lip | DVS Gesture | ASL DVS      |
> | ------------ | ------- | ----------- | ------------ |
> | **Accuracy** | 1.83%   | 35.52±1.35% | 12.65±14.61% |
>
> R2：Comparison of Linear Probing Accuracy with Different Event2vec Methods and with/without Transformer Layers
>
> | Event2vec                                                    | Transformer    | DVS Gesture | ASL DVS      |
> | ------------------------------------------------------------ | -------------- | ----------- | ------------ |
> | None                                                         | None           | 35.52±1.35% | 12.65±14.61% |
> | Parametric Spatial Embedding + Convolutional Temporal Embedding | First 5 layers | 86.94±1.21% | 69.83±7.66%  |
> | Standard Spatial Embedding + Sin Temporal Embedding          | First 5 layers | 84.44±3.35% | 59.71±8.76%  |
> | Parametric Spatial Embedding + Convolutional Temporal Embedding | None           | 56.32±2.56% | 38.90±10.48% |
> | Standard Spatial Embedding + Sin Temporal Embedding          | None           | 37.36±2.22% | 26.63±10.91% |
>
> ## Details of Table 2
> Citations follow Table 1; they were omitted due to space. We compared dataset-specific SOTAs with official open-source code (see Appendix A.6).
> Main differences vs. our method:
> - **Max-Former** (Fang et al., 2025) [DVS Gesture]: converts events to frames; uses a Spiking Transformer/CNN hybrid.
> - **GNN & Transformer** (Yuan et al., 2023) [ASL-DVS]: converts events to frames and voxel graphs; processes via GNNs and Transformers.
> - **Spiking ResNet18 & BiGRU** (Dampfhoffer & Mesquida, 2024) [DVS-Lip]: converts events to frames; uses CNNs and BiGRUs.
>
> In contrast, Event2Vec requires only lightweight sampling or clustering and feeds native vector representations directly into a standard Transformer, fully exploiting GPU parallelism to achieve significantly higher throughput and lower latency than all SOTA competitors.
>
> ## Errors in Eq 5
> Thank you for the clarification. We will fix this error.
>
> ## Reason for showing different methods
> **For the baseline methods:** SOTA methods vary across datasets and use different representations; we selected both SOTA models and diverse representation types for a comprehensive comparison.
>
> **For our proposed method:** Compared with the field of NLP, the datasets we use are relatively small in scale, and our models are also lightweight. Consequently, we are not yet able to train a general-purpose model comparable to BERT or GPT. Therefore, we still adopt different models for different datasets, yet the only differences among them lie in their width and depth.

---

> > ### Author Rebuttal · Reviewer_KJex · 2026-04-05
> >
> > The rebuttal showed the effectiveness of Event2Vec more clearly by providing more direct comparisons of the event representations. This additional discussion raises my confidence in the importance of the Event2Vec representations.
> > I am therefore inclined to raise my rating.

---

> > > ### Author Response · Authors · 2026-04-06
> > >
> > > We sincerely thank you for reviewing our rebuttal and for your highly encouraging feedback. We are thrilled to hear that the additional comparisons clearly demonstrated the effectiveness of Event2Vec and successfully raised your confidence in our work. We noticed your status indicates that you have follow-up questions. We are very eager to address them to fully resolve your remaining concerns and solidify your inclination to raise the rating. Could you please share these follow-up questions with us? We look forward to your further insights and are fully prepared to provide any additional details you may need during this discussion phase.

---

### Decision · Program_Chairs · 2026-04-30

**Decision:**

Accept (regular)

**Comment:**

The authors' rebuttal effectively addressed most concerns. The linear probing experiments in the rebuttal clearly demostrate Event2Vec's standalone effectiveness, showing 56.32% accuracy on DVS Gesture versus 35.52% for raw events and 37.36% for baseline embeddings. The authors proposed Reservoir Sampling for streaming scenarios, acknowledged similarity to Sekikawa CVPR'19 and commited to explicit citation, and explained the temporal difference modeling rationale. Reviewer jKHr confirms "most of my concerns have been addressed." Reviewer wC7c states "the author has resolved most of my questions." Reviewer 3KtX confirms concerns are "adequately addressed." Reviewer KJex noted being "inclined to raise my rating" though had follow-up questions that were not posted. Outstanding issues include evalution on dense prediction tasks (left for future work) and the need to incorporate architectural figures into the main text.

Three reviewers explicitly confirmed their concerns were resolved after rebuttal. Reviewer KJex, the only negative reviewer, indicated being "inclined to raise my rating" based on the rebuttal's additional comparisons, though did not formally update thier score. The rebuttal's linear probing experiments compellingly demonstrate the representation's intrinsic value independent of the Transformer backbone. The AC recommends acceptance, following reviewer majority opinion.

On balance, AC agrees with positive points raised by all reviewers which outweigh the negative points. The authors are strongly encouraged to include the additional reviewer recommendations, experiments from rebuttal and clarifications in the camera-ready version. Specifically, the authors should incorporate Figure 6 from the appendix into the main method section, condense the abstract and remove redundent content in the introduction, explicitly discuss similarity to Sekikawa CVPR'19, fix Equation 5 notation, and include the linear probing comparison tables from the rebuttal. The foundational contribution of enabling sparse event data to be processed by standard Transformers while maintaning asynchronous characteristics represents significant progress for neuromorphic vision research.